# Associations between air pollutants and blood pressure in an ethnically diverse cohort of adolescents in London, England

A. Karamanos[1]*, Y. Lu[1,2], I. S. Mudway[3,4], S. Ayis[5], F. J. Kelly[3,4], S. D. Beevers[3,4], D. Dajnak[3,4], D. Fecht[3,4], C. Elia[1], S. Tandon[5], A. J. Webb[6], A. J. Grande[7], O. R. Molaodi[8], M. J. Maynard[9], J. K. Cruickshank[1], S. Harding[1,5]

1 School of Life Course/Nutritional Sciences, King's College London, London, United Kingdom, 2 Clinical Research Center of The Third Xiangya Hospital, Central South University, Changsha, China, 3 MRC Centre for Environment and Health, School of Public Health, Imperial College London, London, United Kingdom, 4 NIHR Health Protection Research Unit in Environmental Exposures and Health, Imperial College London, London, United Kingdom, 5 Faculty of Life Sciences & Medicine, Department of Population Health Sciences, School of Population Health & Environmental Sciences, King's College London, London, United Kingdom, 6 Faculty of Life Sciences & Medicine, Department of Clinical Pharmacology, King's College London BHF Centre of Excellence, School of Cardiovascular Medicine and Sciences, King's College, London, United Kingdom, 7 Curso de Medicina, Universidade Estadual do Mato Grosso do Sul, Campo Grande, MS, Brazil, 8 MRC/CSO Social and Public Health Sciences Unit, Institute of Health and Wellbeing, University of Glasgow, Glasgow, Scotland, 9 School of Clinical & Applied Sciences, Leeds Beckett University, Leeds, United Kingdom

* alexis.1.karamanos@kcl.ac.uk

**Data Availability Statement:** The DASH data are available to researchers who meet the criteria for access to confidential data via a data request to the MRC/CSO Social and Public Health Sciences Unit,

## Abstract

Longitudinal evidence on the association between air pollution and blood pressure (BP) in adolescence is scarce. We explored this association in an ethnically diverse cohort of schoolchildren. Sex-stratified, linear random-effects modelling was used to examine how modelled residential exposure to annual average nitrogen dioxide ($NO_2$), particulate matter ($PM_{2.5}$, $PM_{10}$) and ozone ($O_3$), measures in μg/m$^3$, associated with blood pressure. Estimates were based on 3,284 adolescents; 80% from ethnic minority groups, recruited from 51 schools, and followed up from 11–13 to 14–16 years old. Ethnic minorities were exposed to higher modelled annual average concentrations of pollution at residential postcode level than their White UK peers. A two-pollutant model ($NO_2$ & $PM_{2.5}$), adjusted for ethnicity, age, anthropometry, and pubertal status, highlighted associations with systolic, but not diastolic BP. A μg/m$^3$ increase in $NO_2$ was associated with a 0.30 mmHg (95% CI 0.18 to 0.40) decrease in systolic BP for girls and 0.19 mmHg (95% CI 0.07 to 0.31) decrease in systolic BP for boys. In contrast, a 1 μg/m$^3$ increase in $PM_{2.5}$ was associated with 1.34 mmHg (95% CI 0.85 to 1.82) increase in systolic BP for girls and 0.57 mmHg (95% CI 0.04 to 1.03) increase in systolic BP for boys. Associations did not vary by ethnicity, body size or socio-economic advantage. Associations were robust to adjustments for noise levels and lung function at 11–13 years. In summary, higher ambient levels of $NO_2$ were associated with lower and $PM_{2.5}$ with higher systolic BP across adolescence, with stronger associations for girls.

University of Glasgow. Application forms and the DASH data sharing policy can be found at http://dash.sphsu.mrc.ac.uk/Data-sharing.html. It reflects the MRC guidance on data sharing with the aim of making the data as widely and freely available as possible while safeguarding the privacy of participants, protecting confidential data, and maintaining the reputation of the study. All potential collaborators work with a link person, an experienced DASH researcher—to support their access to and analysis of the data. The variable-level metadata is available from the study team and via the MRC Data Gateway.

**Funding:** The study was funded by the MRC (10.13039/N4 501100000265, MC_U130015185/ MC_UU_12017/1/ MC_UU_12017/13) North Central London Consortium and the Primary Care Research Network. This work was also supported by the MRC Centre for Environment and Health, which is currently funded by the Medical Research Council (MR/S019669/1, 2019-2024). Infrastructure support for the Department of Epidemiology and Biostatistics was provided by the NIHR Imperial Biomedical Research Centre. The funders had no role in study design, data collection and analysis, decision to publish, or preparation of the manuscript.

**Competing interests:** The authors have declared that no competing interests exist.

## 1. Introduction

Air pollution is a major public health issue, associated with cardio-respiratory disease incidence, hospital admission and deaths [1,2]. Most studies focus on adults, but rapid growth in childhood and adolescence render organ systems particularly susceptible to injury, so that adverse impacts of air pollution in early life may track from childhood with long lasting consequences over the life course [3]. Blood pressure (BP) also tracks from childhood to adulthood; anthropometry, ethnicity and socio-economic disadvantage are recognised influences [4,5]. Whilst the effects of air pollutants on BP in adulthood are well established [6], their impact during the transition from childhood and adolescence are not clear.

In 2022, our literature search identified 24 studies examining air pollutants and BP in children and adolescents, 11 in China [7–17], 1 in Pakistan [18], 3 in the US [19–21] and 9 in Europe [22–30]. Study designs and findings varied. Studies conducted in China were cross-sectional, except one longitudinal study [15], based on large sample sizes of 4–18 year olds, and showed overwhelmingly positive associations between air pollution and BP, either with long-term (equal to a year or more) [7,9,10,12,14,15,17] or short-term exposures (ranging from a few days to several months) [8,11,13,16] to higher levels of air pollutants than recommended by World Health Organization (WHO) and Chinese National Ambient Air Quality Standards (NAAQS) across childhood and adolescence. Positive associations were reported for exposures to ozone ($O_3$) [7,8] and particulate matter ($PM_{2.5}$ and $PM_{10}$) [7,8,14–17], sulphur dioxide ($SO_2$) [7] and nitrogen dioxide ($NO_2$) [7]. Stronger positive associations were observed between PM and BP among those who were underweight [13], overweight [9] or obese [11] compared with those with normal weight. Sex differences varied -with some studies showing stronger associations for females [7,11], males [8–10,13], or no difference [12,14]. The study in Pakistan was a small school based cross-sectional study of 8–12-year-olds and showed a positive association between short-term $PM_{2.5}$ exposure and BP [18]. The three US studies also showed positive associations. Higher maternal long-term $NO_2$ exposures during the third trimester of pregnancy were associated with higher BP at age 11years [20]. Exposures to seven -day average $PM_{2.5}$, $NO_2$, NO, Carbon monoxide (CO) were associated with an increase in oxidative stress, acute inflammation, endothelial dysfunction, and diastolic BP a week later in adolescents aged 14–18 years [19], while another study in 6–8 year-old children in California [21] measuring short term exposures to pollution (1-day lag, and 1-week, 1-month, 3-month, 6-month, and 1-year averages prior to each participant's visit date) found positive associations between $NO_2$ and systolic BP, with the association being strongest at 3 months. A 6.2 ppb increase in 3-month average NO2 was associated with a 9.4 mmHg increase in SBP (95% CI: 2.8, 15.9).

European studies showed mixed results. Three Dutch studies used data from the PIAMA birth cohort, and at 12–16 years, two studies found positive associations with long-term $PM_{2.5}$ and $NO_2$ and diastolic, but not with systolic BP [23,25]; the third found no association [21]. A small Belgium study of 6–12 year olds showed that short-term exposure to ultrafine particles was associated with higher systolic BP [26]. Further a study of 8–12-year-olds in the Tyrol region of Austria and Italy reported that higher annual means of $NO_2$ were associated with lower systolic and diastolic BP [30]. The remaining European studies showed no associations with long-term exposures to air pollutants [22,24,29]. These European adolescents were exposed to lower air pollution to those in China and Pakistan, but in all cases were above the newly revised WHO guideline annual value of 5 mg/m$^3$.

Our Determinants of Adolescent Social Well-Being and Health (DASH) study has followed an ethnically diverse cohort of young people in London, England) since 2002. Given the paucity of studies exploring associations between air pollutants and adolescent BP in the UK, we

tested the air-pollutant-BP hypothesis with high resolution annual average air pollutant ($NO_2$, $O_3$, $PM_{10}$, $PM_{2.5}$) modelled concentration estimates based on an urban dispersion model [31]. We examined BP-air pollutant associations during adolescence and whether these were modified by sex, ethnicity and socioeconomic circumstances (SEC), since evidence shows that ethnic minorities and deprived communities are hardest hit by air pollution in the UK [32], and body mass index (BMI). We hypothesised that long-term exposures to modelled annual average air pollution would be associated with higher BP.

## 2. Methods

### 2.1. Design and sample

The DASH longitudinal study is detailed elsewhere [33]. Briefly, in 2003–2004, 6,631 students, aged 11–13 years, were recruited from 51 secondary schools in 10 London boroughs including Brent, Croydon, Hackney, Hammersmith & Fulham, Haringey, Lambeth, Newham, Southwark, Waltham Forest, and Wandsworth. These boroughs were selected as they have high proportions and numbers of people from ethnic minority groups. Schools with at least 5% of people of Black Caribbean descent were identified using school censuses provided by the Department of Education and Skills. Within each borough, schools were selected to enable representation at, above and below the national averages for academic performance based on reports from the Office for Standards in Education. The classes were randomly selected and were all mixed ability classes. Eighty-three per cent (83%) of eligible students took part in the baseline study. Schools and pupils actively consented to take part in the study. Parents were provided with an information pack and given the opportunity to opt their child out of the study. The baseline sample consisted of approximately 1,000 pupils in each of the 6 main UK ethnic groups. In 2005–2006, 4,775 students took part in follow-up physical measures, a response rate of 88% among those invited. Ethnicity was self-reported, checked against reported parental ethnicity and grandparental country of birth. Ethnic groups were categorised by British 2001 Census categories as: 872 White UK, 713 Black Caribbean, 842 Black African, 397 Indian, and 460 Pakistani and Bangladeshi which were combined for analytical reasons, a total of 3,284. Financial constraints did not allow measurements for participants for the 'Other' ethnic group at follow-up and therefore 1,491 participants from the 'Other' ethnic group were not included in the final analytic sample. Written informed consent was obtained for all participants. Ethics approval was obtained from The Multicentre Research Ethics Committee (MREC) and NHS Local Research Ethics Committees.

### 2.2. Outcome: Blood pressure

Fieldworkers were trained for 1 week prior to the start of fieldwork and were recertified at 6-monthly intervals. Protocols for anthropometric measurements were based on WHO's manual [34]. Systolic and diastolic BP were measured at participants' schools using validated OMRON M5-I instruments, with appropriately sized cuffs, calibrated regularly by field supervisors. First readings were made after pupils sat quietly for a timed 5 minutes, with ≥1 minute between subsequent readings. Averaged second and third readings were analysed.

### 2.3. Exposure: Residential air pollutant assessments

Annual $NO_2$, $O_3$, $PM_{10}$ and $PM_{2.5}$ models measured in micrograms per cubic metre (μg/m$^3$) for London, covering 2003–2004 and 2005–2006 were derived, at a 20m×20m grid point resolution, from the ERG urban dispersion model using the Atmospheric Dispersion Modelling System (ADMS) v4 [31] and road source model v2.3 (Cambridge Environmental Research

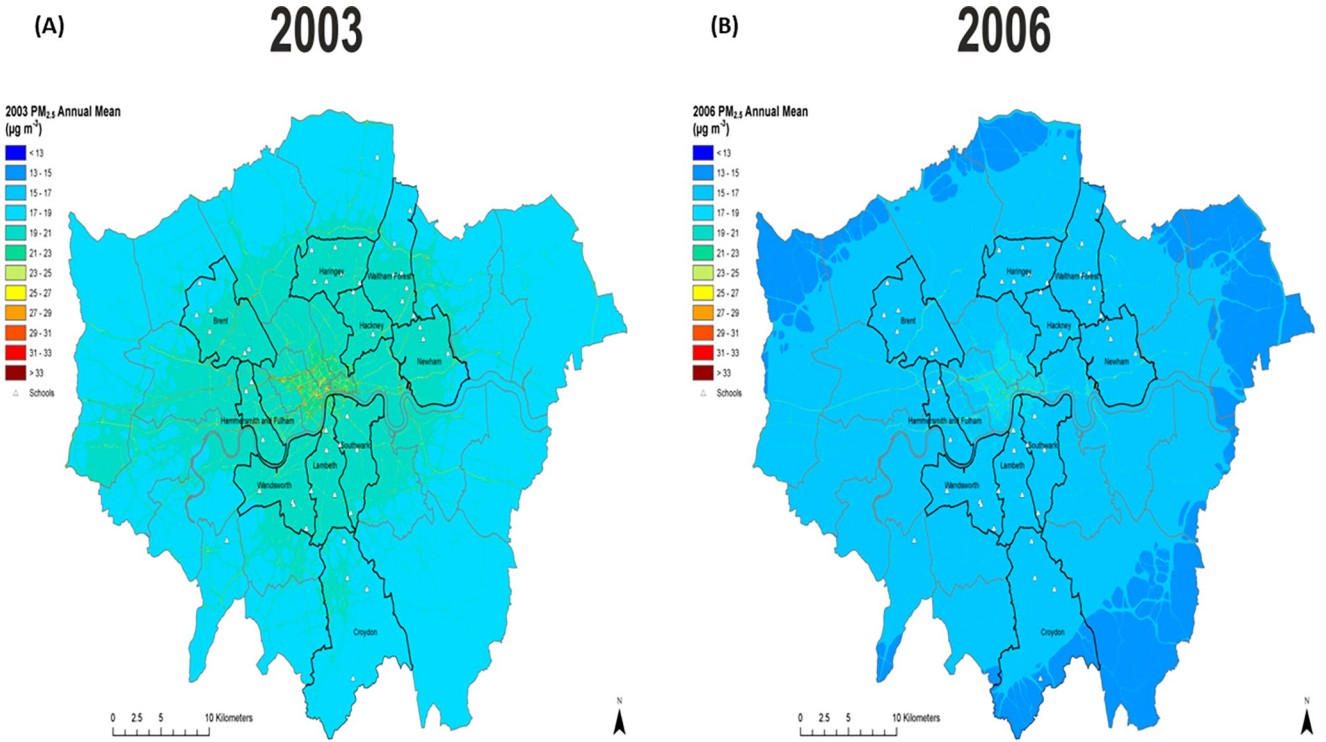

**Fig 1.** Modelled Greater London concentrations ($20m^2$ resolution) for $PM_{2.5}$ for representative years 2003 (A) and 2006 (B). The ten boroughs in which the study took place are highlighted together with the locations of the participating schools (open triangles).

Consultants), hourly measured meteorological data. Empirically derived $NO$-$NO_2$-$O_3$ and PM relationships used emissions from London's Atmospheric Emissions Inventory (see London's pollutant surface maps for 2003 and 2006 as Figs 1 and 2, S1 and S2 Figs). Each yearly model reflected a range of pollutant sources and emissions, including major and minor roads, detailed information on vehicle stock, traffic flows, and speed on a link-by-link basis. Other sources within the model included large and small regulated industrial processes, boiler plants, domestic and commercial combustion sources, agriculture, rail, ships, airports, and pollution carried into the area by prevailing winds. Full details of these models and their validation against measurement data collected within the London Air Quality Network were published recently [35]. Mean daily temperature (˚C) and relative humidity (%) were not considered as previous work in London indicated a similar pattern of correlations between ambient pollution concentrations in warm and cool periods of the year [36,37]. All exposures were estimated using the annual mean for the year in which the assessment was performed within a 20m radius buffer zone around each participant's residential postcode centroid, with fewer than 1% of participants moved homes during the study period.

## 2.4. Covariables

Height was measured using portable stadiometers and weight on Salter electronic scales. Z-scores for body mass index (BMI = weight (kg)/ height ($meters^2$)) and height based on 1990 British children standards [38,39]. Assessment of pubertal status was nurse-supervised using the Tanner stages questionnaire [40]. Prepuberty (Tanner stage 1 for breasts, genitalia, and pubic hair) and early puberty (Tanner stages 2 and 3) were combined, because <5% were

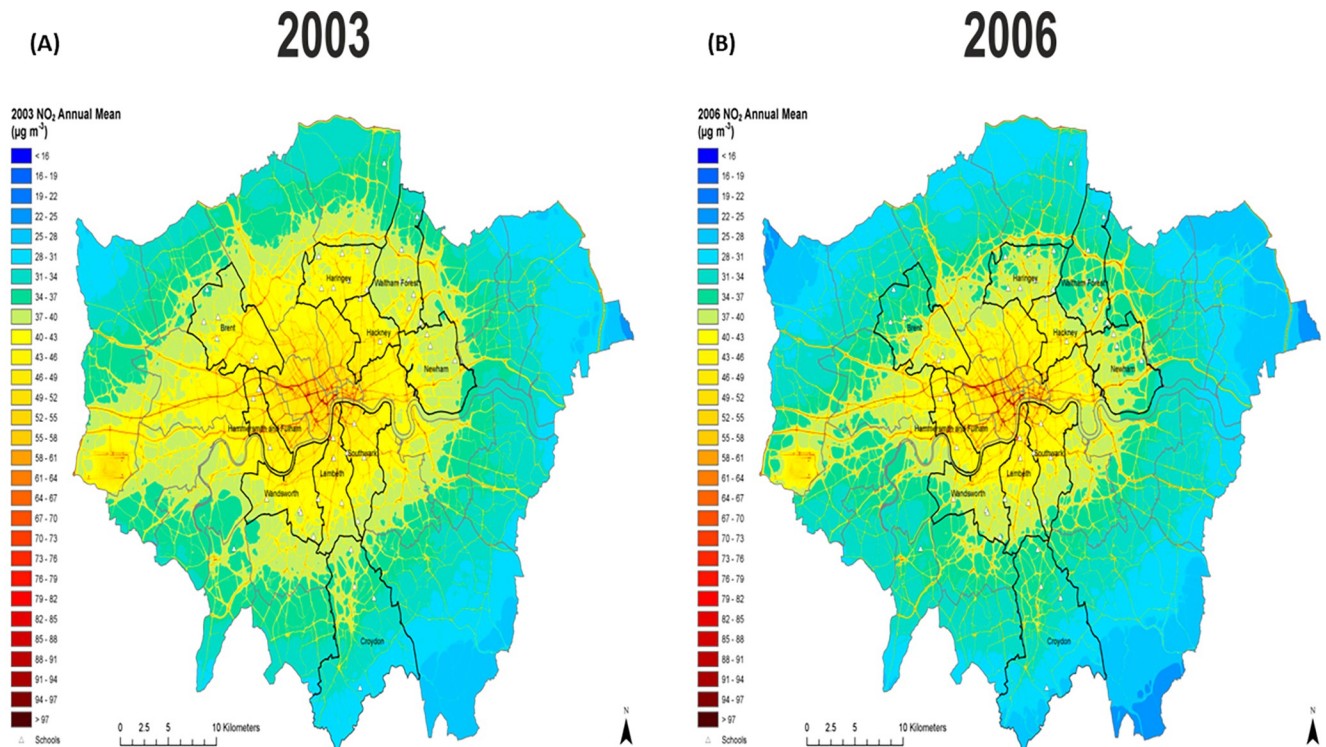

**Fig 2.** Modelled Greater London concentrations (20m$^2$ resolution) for NO$_2$ for representative years 2003 (A) and 2006 (B). The ten boroughs in which the study took place are highlighted together with the locations of the participating schools (open triangles).

classified as prepubertal. Tanner stages 4 and 5 for breasts/ genitalia defined late puberty. Ambient air temperature was recorded where BP was measured.

A self-complete questionnaire was used to collect information on socio-demographics and behaviours. A family affluence scale included family car ownership (0,1,≥2), own bedroom (no = 0, yes = 1), family holidays in the past 12 months (0–3), and family computer(s) (0, 1 2, ≥3). Scale items were combined to a composite score from 0–9 (low to high affluence) [41], then into tertiles. Socio-economic circumstances comprised the family affluence scale score, and a family type measure (2-parent, lone parent) combined with parental employment (yes/ no). Alcohol and tobacco smoking were categorised as ever consumed/smoked, or not. Physical activity was measured as the number of sports activities in the week before the survey and categorized in quartiles. Neighbourhood deprivation was measured by the income domain of the Index of Multiple Deprivation in 2004 and 2007, as quartiles. This index measures the local population proportion receiving income support at the Lower Super Output geographic level, of approximately 1,500 residents or ~650 households.

## 2.5. Statistical analysis

Statistical analyses were conducted using Stata 16 v.1 (Stat Corp., College Station, Texas, USA). P-values <0.05 were considered statistically significant. Systolic/ diastolic BP were not transformed as the Shapiro–Wilk test indicated normal distributions (S3 Fig). We used linear mixed-effects modelling to estimate relationships between air pollutants and BP, adjusted for age in years, age$^2$, z-BMI, z-height, room temperature and pubertal status (Model 1 = core model), then additionally adjusted sequentially for ethnicity, alcohol, smoking, physical activity, family affluence scale, family type/ employment status and neighbourhood deprivation

(Model 2 = full model).Further, we checked for interactions between air pollution and ethnicity, Socio Economic Circumstances (SEC), or BMI by conducting Wald tests. Due to high correlations between different air pollutants (all $rs > 0.9$), a two-pollutant model was also applied while controlling for Model 1 and 2 covariables to adjust the effect estimate of each pollutant for confounding by another. We focused on two weakly correlated pollutants ($PM_{2.5}$ and $NO_2$ (r = 0.4), S4 Fig) i) for statistical parsimony; ii) because these two pollutants present significant regulatory challenges in Britain, meeting the European Union's (EU) $NO_2$ annual standard and WHO's guideline for $PM_{2.5}$, as the stated aim of current policy, and iii) since limiting analyses to $NO_2$ and $PM_{2.5}$ is efficient for dealing with general sources defined as homogeneous and regional ($PM_{2.5}$), versus a spatial variable reflecting primary combustion ($NO_2$).

A comparison between participants with observed/complete and participants with missing data in one or more analyses variables highlighted systematic differences as those with missing data were more likely to be from an ethnic minority and more socio-economically deprived background (S1 Table). Therefore, the Multiple Imputation by Chained Equations (MICE) approach was used to handle missing data. BP data was missing for <4% of the sample but 26% had missing pollutant data from incomplete residential postcodes; 56% of the sample had complete data in all analyses. Overall, fifty imputations were generated under the 'Missing-at-Random' assumption; distributions of complete and imputed variables were checked [42]. The imputation model was stratified by sex since statistical interactions added to a complete case fully adjusted Model (indicated significantly different slopes for boys and girls ($NO_2$, -0.37, 95% CI -0.57 to -0.17, p<0.001 and $PM_{2.5}$, 1.9, 95% CI 1.6 to 2.3, p<0.001). The imputation model included all exposures, covariables as listed above for Model 2, covariables for sensitivity checks (levels during day, evening, and night noise (Lden) and spirometry at 11–13 years), the outcome of interest (systolic/ diastolic BP), and auxiliary variables (generational status from country of birth (abroad or UK) to help predict missing data. Interaction terms between ambient air pollutants, ethnicity, family affluence scale tertiles and BMI were included. Estimates were combined using Rubin's rules [43]. Descriptive results from the imputed data are shown in Table 1 and from complete case data in S2 Table, while imputed values on BP (n = 243) were dropped when applying sex-stratified multivariable linear random effects modelling to protect estimates from possible problematic imputations [44], with 1,630 observations for boys and 1,409 for girls being finally retained.

## 2.6. Sensitivity analyses

To test whether analyses for the main findings change upon considering further assumptions, a series of sensitivity analyses were conducted. First, we replicated these analyses in participants with 'complete' data in the exposures, covariates, and BP. Second, to check the influence of very low and very high annual modelled pollutant concentrations of air pollution, we restricted exposures to the 1st to 99th percentile. Third, to explore potential confounding from road-traffic noise, 24-hour annual average noise levels with 10 decibels (db) penalty for night-time noise and 5db penalty for evening noise ($L_{den}$) at participants' residential postcode centroid were modelled, using the TRANEX model [45]. Fourth, since lung function and BP development were associated in DASH previously [46] and lung capacity is linked to air pollution levels [35], we explored whether lung volume at 11-13y confounded the associations of interest in Model 2. Lung function testing used portable Micro Plus spirometers (Micro Medical Ltd., Kent, UK) (based on American Thoracic and European Respiratory Society guidelines [47]. Forced Expiratory Volume$_1$ ($FEV_1$) and Forced Vital Capacity (FEV) were recorded for each manoeuvre. The best $FEV_1$ and FVC for each child were used in analysis. Due to strong correlation between $FEV_1$ and FVC in DASH (r = 0.8) [48], separate models were run.

**Table 1. Selected descriptive characteristics of the DASH participants (multiple imputed data)[±].**

| | Boys 11–13 years (n = 1773) | Girls 11–13 years (n = 1511) | Boys 14–16 years (n = 1773) | Girls 14–16 years (n = 1511) |
|---|---|---|---|---|
| **Blood Pressure** | | | | |
| Mean systolic (95% CI) | 108.6 (108.5–108.7) | 108.6 (108.5 to108.7) | 114.5 (114.4–114.5) | 107.8 (107.8–107.8) |
| % > or = 90th Centile[§] (95% CI) | 4.0 (3.1 to 5.0) | 5.6 (4.5 to 6.8) | 11.1 (9.6 to 12.6) | 4.1 (3.1 to 5.1) |
| Mean diastolic (95% CI) | 66.1 (66.1–66.2) | 67.9 (67.9 to67.9) | 68.4 (68.3–68.5) | 68.8 (68.7–8.8) |
| % > or = 90th Centile[§](95% CI) | 4.1 (3.2 to 5.0) | 5.0 (3.9 to 6.1) | 6.7 (5.5 to 7.9) | 6.2 (4.9 to 7.4) |
| **Elevated systolic/diastolic Blood pressure** | | | | |
| % > or = 90th Centile[§] (95% CI) | 6.7 (5.5 to 6.7) | 8.7 (7.3 to 10.1) | 14.5 (12.8 to 16.2) | 8.4 (7.0 to 9.8) |
| **Air Pollutants** | | | | |
| | **Median (IQR)** | **Median (IQR)** | **Median (IQR)** | **Median (IQR)** |
| $NO_2$(µg/m$^3$) | 41.3 (39.0 to-43.5) | 41.4 (39.1 to 43.7) | 40.7 (38.3 to43.0) | 40.4 (37.8 to 43.2) |
| $PM_{2.5}$ (µg/m$^3$) | 19.4 (19.0 to-19.7) | 19.4 (19.0 to19.7) | 16.1 (15.7 to-16.4) | 16.1 (15.7- to 16.4) |
| $PM_{10}$ (µg/m$^3$) | 28.9 (28.3–29.4 to) | 28.9 (28.3 to-29.49) | 25.1 (24.4 to25.4) | 25.0 (24.2 to-25.6 to) |
| $O_3$ (µg/m$^3$) | 34.1 (32.5 to35.5) | 34.0 (32.4 to-35.5) | 36.4 (34.9 to 37.7) | 36.7 (34.7 to-38.6) |
| | **Mean % (95% CI)** | | | |
| **Family Affluence Score[+]** | | | | |
| Least disadvantaged (Highest tertile) | 22.2 (20.1 to24.3) | 17.5 (15.5 to19.5) | 26.4 (24. to28.6) | 23.5 (21.3 to25.8) |
| Most disadvantaged (Lowest tertile) | 58.1 (55.6 to60.6) | 40.5 (37.8 to43.1) | 52.9 (50.5 to55.4) | 32.9 (30.4 to35.6) |
| **Family type+** | | | | |
| 2-parent family, > = 1 employed | 66.3 (66.0 to66.6) | 61.3 (61.0 to61.6) | 66.1 (65.9 to66.4) | 61.1 (60.8 to61.4) |
| lone-parent family, 0 employed | 10.8 (10.7 to11.0) | 12.5 (12.3 to12.7) | 7.1 (6.9 to7.2) | 7.4 (7.1 to7.6) |
| **Physical activity (number of activities) [+]** | | | | |
| Highest quartile | 31.0 (29.6 to33.4) | 15.1 (13.2 to17.0) | 28.5 (26.4 to30.7) | 19.3 (17.3 to21.4) |
| Lowest quartile | 15.9 (14.1 to7.7) | 35.6 (33.1 to38.2) | 23.3 (21.3 to25.4) | 32.3 (29.8- to 34.7) |
| **Alcohol intake (%Yes)** | 30.7 (30.4 to31.0) | 31.5 (31.2 to31.8) | 44.0 (43.8 to44.4) | 54.6 (54.3 to54.7) |
| **Tobacco smoking (%Yes)** | 20.7 (18.6 to22.7) | 19.3 (17.2 to21.4) | 38.4 (36.0 to40.7) | 46.4 (43.8 to49.0) |
| **Pubertal stage (Late)** | 45.5 (43.0 to48.2) | 53.8 (51.1 to56.6) | 90.0 (88.9 to91.7) | 87.9 (86.2 to89.6) |
| **IMD-Income domain)[†+]** | | | | |
| Least deprived quartile | 31.5 (29.3 to33.7) | 30.2 (27.8 to32.5) | 22.4 (20.4 to24.4) | 22.1 (19.9 to24.2) |
| Most deprived quartile | 18.9 (17.0 to0.8) | 22.8 (20.6 to24.9) | 27.6 (25.5 to29.8) | 30.5 (28.1 to32.9) |

[§]Height-percentile–specific BP at ≥90th percentile levels were classified as elevated BP using European and US normative BP tables for children and adolescents [49,50].

[±] Selected descriptive characteristics of the DASH participants using complete cases are shown in S2 Table.

[+] Descriptive statistics for all categories are shown in S3 Table.

[†]Index of Multiple Deprivation.

IQR-Interquartile Range.

## 3. Results

### 3.1. Descriptive findings

At age 11–13 years, the mean systolic BP was similar in boys and girls (Table 1), while by age 14–16 years, was higher in boys than girls (115vs 108mmHg). At age 11–13 years, girls had slightly higher elevated BP (% > or = 90th Centile according to European [49] and US normative BP tables [50] than boys at 11–13 years (8.7% vs. 6.7%), although not statistically significant. However, at age 14–16 years, boys were almost two times as likely to have elevated BP as girls (14.5% vs. 8.4%), which were due to elevated levels of systolic BP. Boys were more likely to have ever smoked and drank alcohol, and engaged more frequently in physical activity than

**Table 2. Modelled annual exposures to median $O_3$, $NO_2$, $PM_{2.5}$ and $PM_{10}$ concentrations (F06Dg/m[F033]) by ethnic group during the study's follow-up period.**

| | Boys | | | | |
|---|---|---|---|---|---|
| | **White UK (IQR)** | **Black Caribbean (IQR)** | **Black African (IQR)** | **Indian (IQR)** | **Pakistani & Bangladeshi (IQR)** |
| | (n = 487) | (n = 355) | (n = 389) | (n = 225) | (n = 317) |
| Pollutant | | | | | |
| $O_3$ ($\mu g/m^3$) | 35.9 (34.0 to 37.6) | 35.1 (33.4 to 37.0) | 34.1 (32.3 to 35.9) | 35.8 (34.6–37.1) | 35.5 (34.4 to 36.9) |
| $NO2$ ($\mu g/m^3$) | 40.0 (37.8–42.4) | 41.2 (38.8 to 43.5) | 42.9 (39.9 to 45.2) | 40.1 (38.5 to 41.0) | 40.7 (38.9 to 42.0) |
| $PM_{2.5}$ ($\mu g/m^3$) | 17.6 (15.9–19.2) | 17.7 (16.0–19.3) | 18.0 (16.3 to 19.6) | 17.6 (15.9 to 19.2) | 17.7 (15.9 to 19.2 |
| $PM_{10}$ ($\mu g/m^3$) | 26.7 (24.7–28.5) | 27.0 (24.9 to 28.9) | 27.4 (25.3 to 29.2) | 26.8 (24.7 to 28.5) | 27.0 (24.9 to 28.6) |
| | Girls | | | | |
| | **White UK (IQR)** | **Black Caribbean (IQR)** | **Black African (IQR)** | **Indian (IQR)** | **Pakistani & Bangladeshi (IQR)** |
| | (n = 385) | (n = 358) | (n = 453) | (n = 172) | (n = 143) |
| Pollutant | | | | | |
| $O_3$ ($\mu g/m^3$) | 36.4 (34.3 to 38.5) | 34.8 (32.7–36.7) | 34.8 (32.6 to 36.5) | 35.9 (34.6 to 37.2) | 35.4 (34.2 to 36.8) |
| $NO_2$ ($\mu g/m^3$) | 39.3 (37.0 to 41.5) | 41.9 (39.2 to 44.2) | 41.9 (39.3 to 44.4) | 39.9 (38.2 to 40.9) | 40.8 (38.7 to 41.6) |
| $PM_{2.5}$ ($\mu g/m^3$) | 17.5 (15.8 to 19.0) | 17.8 (16.1 to 19.5) | 17.9 (16.1 to 19.5) | 17.6 (15.8 to 19.2) | 17.7 (15.9 to 19.2) |
| $PM_{10}$ ($\mu g/m^3$) | 26.5 (24.4 to 28.4) | 27.1 (25.0 to 29.0) | 27.2 (25.1 to 29.1) | 26.7 (24.6 to 28.5) | 26.9 (24.8 to 28.5) |

IQR-Interquartile Range.

girls; compared with White UK children, Black Caribbean and Black African children were also more likely to live in deprived inner-city areas in DASH [51].

Between 2003 to 2006 annual average concentrations of $PM_{10}$ in London fell (Figs 1, S1 and S2), with a similar trend observed for $PM_{2.5}$ (Fig 1), though concentrations remained above the WHO (World Health Organization) annual guideline of 5 mg/m$^3$ for the full duration of the study. The pattern for annual $NO_2$ was less clear cut with decreased concentrations in the outer London (Fig 1), but with slight change in the inner-city boroughs and evidence of increased concentrations at roadside monitoring sites (S2 Fig). Over the study period 54% of the children lived at addresses where annual exposures to $NO_2$ were above the EU limit value for $NO_2$ (40 μg/m$^3$) and well above the WHO guideline values of 10 mg/m$^3$. Despite these changes in overall annual concentrations these patterns did not translate to statistically significant differences when exposure was considered at residential address level (Table 1) when participants were aged 11–13 and 14–16 years of age.

During the follow-up period, Black Caribbean and Black African girls and boys were exposed to higher concentrations of PM and $NO_2$ than their White peers (Table 2). In contrast, White UK and Indian participants were exposed to higher concentrations of $O_3$ compared to their Black Caribbean, Black African and Pakistani and Bangladeshi peers. This is mainly due to the inverse relationship between $O_3$ and $NO_2$, but it also reflects the distribution of ethnic groups throughout London, particularly the large Indian community in outer West London.

## 3.2. Longitudinal relationships between pollutants and BP

Associations between systolic and diastolic BP and modelled annual average pollutant concentrations are summarised in Table 3, based on single and two-pollutant models, the latter focusing on $NO_2$ and $PM_{2.5}$ only. In the single pollutant model, among boys $O_3$ was positively and $NO_2$ negatively associated with systolic BP, and among girls $NO_2$ was negatively associated with both systolic and diastolic BP and $PM_{2.5}$ positively associated with systolic BP. The two-pollutant model showed that $NO_2$ and $PM_{2.5}$ were both significantly associated with systolic BP, but in opposite directions. $NO_2$ was significantly associated with lower systolic BP (b

**Table 3. Longitudinal effects of air pollutants on blood pressure among participants aged ll-16 years in the DASH study (multiple imputed data).**

| | | Boys | | | | Girls | | | |
|---|---|---|---|---|---|---|---|---|---|
| | | Single-Pollutant Model | | Two-pollutant Model | | Single-Pollutant Model | | Two-pollutant Model | |
| **Pollutants** | | | | | | | | | |
| **Systolic BP** | | β (95% CI) | P>|z| | β (95% CI) | P>|z| | β (95% CI) | P>|z| | β (95% CI) | P>|z| |
| $O_3$ (μg/m$^3$) | Model 1 | 0.19 (0.05 to 0.34) | 0.01 | - | - | 0.09 (-0.04 to 0.22) | 0.162 | - | - |
| | Model 2 | 0.21 (0.06 to 0.36) | 0.007 | - | - | 0.10 (-0.03 to 0.24) | 0.138 | - | - |
| $NO_2$ (μg/m$^3$) | Model 1 | -0.11 (-0.20 to -0.02) | 0.017 | -0.19 (-0.30 to -0.08) | 0.001 | -0.11 (-0.20 to -0.03) | 0.007 | -0.28 (-0.38 to -0.17) | <0.001 |
| | Model 2 | -0.12 (-0.22 to -0.02) | 0.014 | -0.19 (-0.31 to -0.07) | 0.002 | -0.13 (-0.22 to -0.04) | 0.004 | -0.30 (-0.40 to -0.18) | <0.001 |
| $PM_{2.5}$ (μg/m$^3$) | Model 1 | 0.02 (-0.38 to 0.42) | 0.911 | 0.54 (0.04 to 1.03) | 0.034 | 0.48 (0.09 to 0.87) | 0.017 | 1.31 (0.83 to 1.79) | <0.001 |
| | Model 2 | 0.04 (-0.37 to 0.46) | 0.834 | 0.57 (0.06 to 1.07) | 0.027 | 0.51 (0.11 to 0.91) | 0.013 | 1.34 (0.85 to 1.82) | <0.001 |
| $PM_{10}$ (μg/m$^3$) | Model 1 | -0.14 (-0.42 to 0.13) | 0.301 | | | 0.10 (-0.16 to 0.37) | 0.436 | - | - |
| | Model 2 | -0.15 (-0.43 to 0.14) | 0.314 | | | 0.12 (-0.16 to 0.39) | 0.402 | - | - |
| **Diastolic BP** | | β | P>|z| | β (95% CI) | P>|z| | β | P>|z| | β (95% CI) | P>|z| |
| $O_3$ (μg/m$^3$) | Model 1 | 0.04 (-0.07 to 0.14) | 0.488 | - | - | 0.06 (-0.04 to 0.15) | 0.236 | - | - |
| | Model 2 | 0.03 (-0.08 to 0.14) | 0.57 | - | - | 0.04 (-0.06 to 0.14) | 0.438 | - | - |
| $NO_2$ (μg/m$^3$) | Model 1 | -0.03 (-0.10 to 0.03) | 0.352 | -0.07 (-0.15 to 0.02) | 0.11 | -0.07 (-0.13 to -0.01) | 0.033 | -0.10 (-0.17 to -0.02) | 0.018 |
| | Model 2 | -0.03 (-0.10 to 0.04) | 0.417 | -0.07 (-0.16 to 0.02) | 0.114 | -0.06 (-0.12 to 0.01) | 0.096 | -0.09 (-0.17 to 0.00) | 0.039 |
| $PM_{2.5}$ (μg/m$^3$) | Model 1 | 0.08 (-0.21 to 0.37) | 0.589 | 0.27 (-0.08 to 0.64) | 0.136 | -0.07 (-0.37 to 0.22) | 0.628 | 0.21 (-0.16 to 0.58) | 0.26 |
| | Model 2 | 0.10 (-0.20 to 0.40) | 0.505 | 0.27 (-0.09 to 0.63) | 0.145 | -0.01 (-0.31 to 0.29) | 0.935 | 0.22 (-0.15 to 0.59) | 0.237 |
| $PM_{10}$ (μg/m$^3$) | Model 1 | -0.01 (-0.20 to 0.19) | 0.957 | - | - | -0.12 (-0.31 to 0.08) | 0.247 | - | - |
| | Model 2 | -0.00 (-0.21 to 0.20) | 0.996 | - | - | -0.08 (-0.28 to 0.12) | 0.438 | - | - |

Model 1: Coefficients were estimated with random effects models, adjusted for age, age$^2$ zBMI, zHeight, ambient air temperature and pubertal stage.

Model 2: Model1+ ethnicity, alcohol, smoking, physical activity, family type and parental employment, family affluence score and neighbourhood deprivation.

coefficient: -0.19; 95% CI -0.30 to -0.08 and b: -0.28; 95% CI -0.38 to -0.17 mmHg per unit (μg/m$^3$) increase in pollutant concentration) in boys and girls respectively, after adjustment for age, BMI, height, ambient air temperature and pubertal stage–Model 1 (see Table 3). By contrast, $PM_{2.5}$ exposures were associated with higher systolic BP of b:0.54; 95% CI 0.04 to 1.03 (boys) and b:1.31; 95% CI 0.83 to 1.79 (girls). Smaller changes of similar directionality were seen with diastolic BP, with only the negative association with annual $NO_2$ exposures in girls attaining statistical significance: b: -0.09; 95% CI-0.16 to -0.02, mmHg per μg/m$^3$ increase, p = 0.01 (S3 Table). Further adjustments for a range of social economic and lifestyle factors (Model 2) did not change the coefficients substantially. Estimations were robust when mean concentrations of $NO_2$ were restricted to 30–55 μg/m$^3$ which correspond to the range from 1[st] to 99[th] percentile, with systolic BP decreasing by 5mmHg (95% CI 2 to 8) for boys and a larger (but overlapping) effect of 8 mmHg (5 to 10) for girls (Fig 3). When restricting mean $PM_{2.5}$ levels to 15–21 μg/m$^3$ (1[st] to 99[th] percentile), systolic BP increased by 3mmHg (95% CI 0.2 to 6) for boys and 8mmHg (5 to 11) for girls. Lack of statistically significant interactions between pollutants, ethnicity, and BMI (F values >3, p >0.1, except for P.M$_{2.5}$ x SEC, F = 3.55, p = 0.06) suggested that pollutant effects in Table 2 and Fig 3 did not vary by ethnicity, BMI, and SEC for either boys or girls.

### 3.3. Sensitivity analyses findings

Comparison of results obtained from multiple imputed and complete case multivariable analyses did not show substantial differences (S4 Table). Restricting exposures to the 1[st] to 99[th] percentile did not change the associations of interest. Similarly, taking into consideration 24-hour

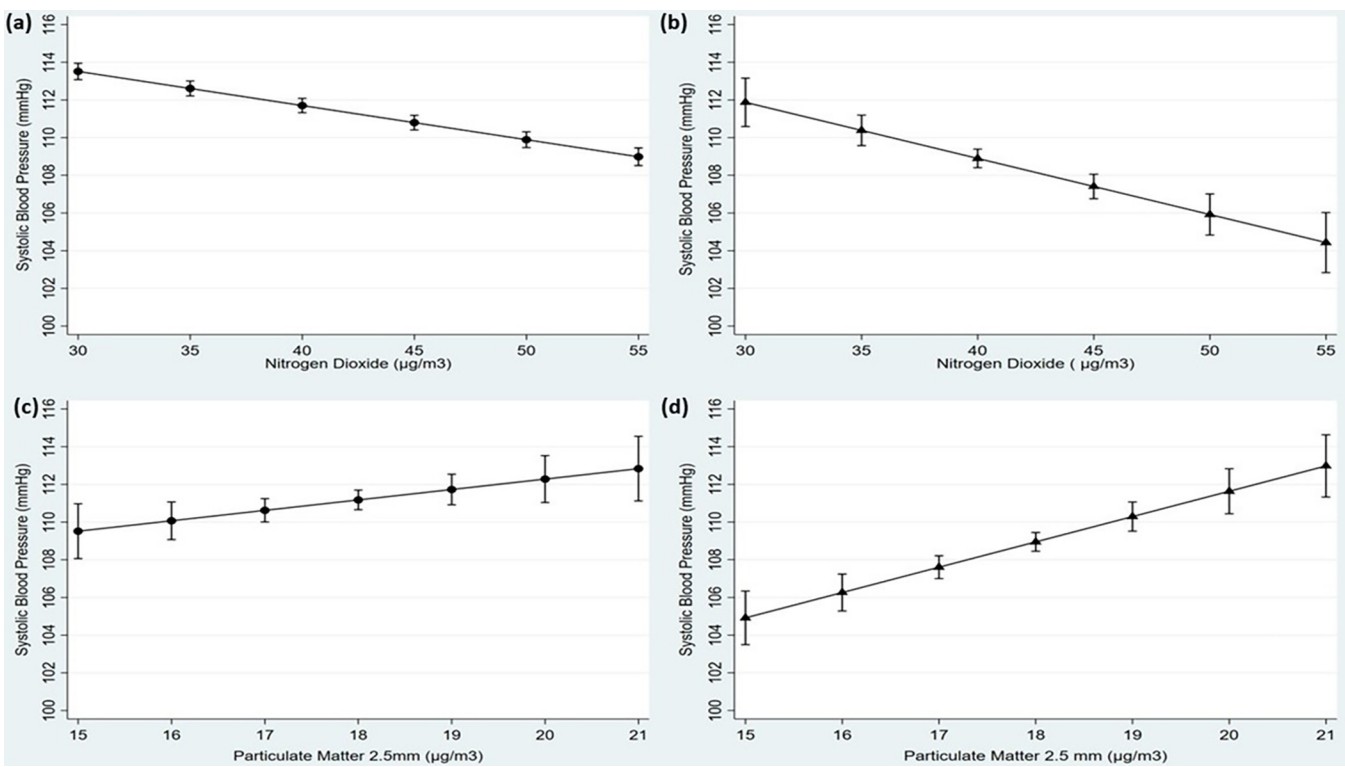

**Fig 3.** 95% confidence intervals of systolic blood pressure by $NO_2$ and $PM_{2.5}$ in boys (panels a & c) and girls (b & d) using data from the DASH study. Predicted means were estimated from random effects models adjusted for the combined effect of $NO_2$ and $PM_{2.5}$, age, age$^2$, ethnicity, anthropometry (z scores of BMI, height) pubertal stage, ambient air temperature, family type and parental employment, smoking, alcohol consumption, smoking, physical activity, and neighbourhood deprivation (Model 2). Means were restricted to $NO_2$ 30–55 μg/m$^3$ and $PM_{2.5}$ 15–21 μg/m$^3$ where estimations were robust (1st to 99th percentile).

noise levels did not alter the associations between systolic BP and $NO_2$ (b = -0.19, 95% CI -0.31 to -0.07, p = 0.003 for boys; b: -0.27, 95% CI -0.39 to -0.16, p <0.001 for girls) and $PM_{2.5}$ (b:0.57, 95% CI 0.07 to 1.07, p = 0.027 for boys; b:1.38, 95% CI 0.89 to 1.87, p <0.001). Adjustment for $FEV_1$ at 11-13y reduced the magnitude of the association between PM and systolic BP for boys (a coefficient reduction to b:0.10, 95% CI -0.44 to 0.64, p = 0.705) while adjustment for $FEV_1$ reduced slightly the magnitude of the association between $NO_2$ and systolic BP (a coefficient reduction to b: -0.14, 95% CI -0.26 to -0.02, p = 0.019). Adjustment for FVC resulted in a lower reduction in the coefficients of interest (a coefficient reduction to b:0.39, 95% CI 0.12 to 0.91, p = 0.137 for $PM_{2.5}$, and a reduction to b: -0.18, 95% CI -0.29 to -0.06, p = 0.004 for $NO_2$). For girls, statistical control for $FEV_1$ reduced the magnitude of the association between $PM_{2.5}$ and systolic BP (a coefficient reduction to b:0.93, 95% CI 0.40 to 1.46, p<0.001,) and between $NO_2$ and systolic BP (a reduction to b:-0.27, 95% CI -0.38 to -0.16, p<0.001). A smaller reduction in the associations of interest was observed after controlling for FVC (a reduction to b:1.17, 95% CI 0.63 to 1.68, p<0.001 for $PM_{2.5}$ and no change for $NO_2$).

## 4. Discussion

This longitudinal study in London, England, indicates that air pollution appears to have a considerable impact on BP during adolescence, and in girls more than boys. We hypothesised that long-term exposure to poor air quality would be associated with higher BP, but found a mixed picture, varying in direction by pollutant. Associations were significantly modified by sex.

Systolic BP was lower in both boys and girls with increasing modelled annual average $NO_2$ concentrations, which in London is predominately due to diesel traffic, whereas the association between elevated $PM_{2.5}$ and higher systolic BP was greater in girls than in boys. Ethnic-specific associations were not evident despite cumulative, marginally higher exposures to modelled annual average $NO_2$ and PM concentrations among Black Caribbeans, Black Africans and Pakistani/ Bangladeshis than their White UK peers. Associations did not vary by SEC or BMI.

A negative $NO_2$-BP relationship can be biologically plausible as we have shown that $NO_2$ feeds into the (nitrate)-nitrite-nitric oxide (NO) pathway and acutely lowers BP [52,53]. In an acute, randomized, controlled, crossover study in 12 healthy participants (aged 26±4years, 10/ 12 female, SBP 113.8±7.9 mm Hg, DBP 72.8±5.7 mm Hg), we found that 90 minutes exposure to $NO_2$ (sitting next to a domestic gas cooker with gas hobs lit and uncovered) versus control (room air) acutely increased plasma [nitrite] and decreased blood pressure by ~5/5 mmHg [52]. This rapid increase in plasma [nitrite] suggested chemical conversion from $NO_2$ (eg, via a nitrous acid intermediary), through a novel Eco physiological NOx cycle, resulting in intravascular NO formation inducing vasodilation with BP-lowering. This process is distinct from the PM-mediated induction of inflammatory pathways [54], but consistent with our findings with sources of dietary nitrate (e.g. beetroot juice), which, via the nitrate-nitrite-NO pathway lowers BP both acutely [55] and chronically over 6-months [56,57]. Apart from the inflammatory responses, exposure to higher PM concentrations may include disrupted circadian rhythms of renal sodium handling, as exposure to higher PM concentrations may reduce the ability of the kidney to excrete sodium during the daytime, leading to a higher night-time BP level [58].

## 4.1. Comparison with other studies

In tandem with previous literature, this study highlighted higher BP levels with exposure to higher concentrations of PM. Our findings are inconsistent with previous studies showing positive associations between $NO_2$ and BP in a Chinese [7] and three US based studies in California [19–21], while it partly corroborates the findings of a study of schoolchildren in the Tyrol area in Austria and Italy which found lower BP with exposure to higher concentrations of $NO_2$ [30]. Stronger associations between PM and systolic BP were observed which is inconsistent with other studies highlighting stronger associations with diastolic BP [23,25] or equally strong associations with both systolic and diastolic BP [7,8,14–19]. In the limited literature on pollution effects in adolescence, there is little analysis of sex differences. Not only did we find that girls by follow-up aged 14–16 years have near half the prevalence rate of 'elevated BP' as boys, but the associations between $NO_2$, $PM_{2.5}$ and systolic BP was greater for girls. The data do not allow us to determine whether differential susceptibility and/or exposure at a micro-residential level could explain the findings, but sex specificity is feasible, owing to differences in the stage of lung development and hence BP [46]. Differences in sporting activities could clearly be relevant as boys had twice the proportion in the high physical activity time quartile at baseline (31%) than girls (15%) (see Table 1). That activity difference persisted, but less so, at follow-up (28.5% vs 19.3%), but perhaps more important, still 30% of girls at 14–16 years were in the lowest activity quartile. It is thus imperative that air pollution is improved in London to maximise the health benefits of physical exercise in young people. Only two Chinese studies also showed stronger PM-BP associations in girls than boys [7,11], however, pollutant concentrations and the method of exposure assessment differed from the current study.

Without longitudinal data, pollutant effects over time cannot be studied, but how different age groups are affected can be. Nevertheless, a widespread preventable increase as small as 1 mmHg in usual systolic BP can increase CVD deaths by 2–4% [59] which cautions against

complacency. In 2019, systolic BP was the leading Level 2 risk factor among 19 other (including air pollution, child and maternal malnutrition, and high BMI) Level 2 risk factors globally for attributable deaths, which accounted for 10.8 million (95% CI 9.51–12.1) deaths or 19.2% (95% 16.9 to 21.3) of all deaths in 2019 [60]. Plausible pathways underlying an effect of particulate air pollution on BP include impaired vascular endothelial function mediated by oxidative stress and inflammation [61–64], and a stimulation of airway nerve endings directly to alter autonomic reflexes. Furthermore, results from recent studies indicate that epigenetic effects via *Alu* and *TLR4* hypomethylation may represent a novel mechanism mediating environmental BP effects [65].

## 4.2. Strengths and weaknesses

DASH has had high retention rates and low item-non-response, due to enormous community support [33]. The cohort's composition is unique with purposely large numbers from major ethnic groups, careful observer training, and analysis. It is plausible that exposure to modelled annual average pollutant concentrations may not be representative of the adolescent population aged 11–16 years in London because of the purposive (non-probability) selection of the DASH sample. Nevertheless, it is likely that the associations between annual modelled annual average pollutant concentrations and BP may generalise to adolescents in urban settings in the UK [66]. Boys had a higher proportion of missing data in one or more analyses variables which may account for the stronger pollutant-BP associations in girls. Although our best efforts to address item non-response bias through multivariable multiple imputations, it remains possible that our imputation model has been miss specified. Other weaknesses include lack of statistical adjustment for atopic diseases which have been previously found to be associated with air pollution and cardiovascular outcomes, as well as the lack of data before age 11 years. It therefore misses air pollution exposure during earlier childhood, or data on birthweight and maternal exposure to air pollution. These factors might impact BP tracking throughout childhood, adolescence and into adulthood. BP was measured on a single occasion, 3 times at each survey, and a more precise estimate would have been obtained from multiple measures over several visits. Further, the availability of lung function measures only at 11–13 years did not allow us to explore whether changes in lung function over time fully mediated the associations between the air pollutants and BP in adolescence. There have been reductions in PM concentrations in London since 2006 and the overall pattern of changes has been published previously [35]. Until about 2012 $NO_2$ concentrations remained relatively static over time within the city, but because of the introduction of London's Low Emission and Ultra Low Emission Zones concentrations have fallen, though not at a sufficient rate to meet EU Limit values in the immediate future. Air pollution in London therefore remains high and well above the WHO air pollution guidelines for both $NO_2$ and $PM_{2.5}$. In the light of this study's findings, it would be important future studies adopt a life course approach and examine the relationship between BP changes and changes in pollutant concentrations in London over the periods of the introduction of Clear Air Zones and reductions in traffic during the COVID-19 lockdowns. Particularly, the period of COVID-19 lockdowns serves as an opportunity to address the independent impacts of $NO_2$ and $PM_{2.5}$ on cardiovascular function due to a substantial change in the ratio between these pollutants (a decrease in $NO_2$ in tandem with relatively static $PM_{2.5}$ levels). Other data limitations include the lack of exposure to the concentration of air pollutants from other locations where adolescents spend much of their time (e.g. school), the absence of time-activity patterns and potentially other environmental confounding such as lack of exposure to green space regarding the positive association between PM2.5 and BP.

### 4.3. Conclusion

Associations between air pollutants and BP vary by sex in adolescence. Further longitudinal studies are needed to clarify the contrasting associations of ambient concentrations of $NO_2$ versus fine particulate matter and BP levels in young people from different socio-economic backgrounds. Understanding the social and biological mechanisms linking air pollution exposure to BP over the life course is major research and clinical gap [67].

## Supporting information

**S1 Fig. Modelled Greater London concentrations (20m2 resolution) for O3 and PM10 for representative years 2003 and 2006.** The ten boroughs in which the study took place are highlighted together with the locations of the participating schools (open triangles). (DOCX)

**S2 Fig. Changes in monthly average concentrations of NO2, PM10 and O3 between 2002 and 2008 using sites classified as inner and outer London roadside (RS) and background (BG) locations.** For $NO_2$ the data from the following number of sites were averaged over the period: Inner London RS (n = 6), outer London RS (n = 7), inner London BG (n = 4), outer London BG (n = 9). For PM10 the equivalent site number were n = 7, 10, 3 and 5, and for O3: n = 3, 3, 4 and 5. Insufficient sites monitored PM2.5 at the beginning of the study period and therefore trends are not shown. In each panel the rate of change in pollutant concentration is illustrated (mean with 95% confidence interval), as μg/m3 per year'. (DOCX)

**S3 Fig. Blood pressure distributions by sex and DASH sweep (vertical lines highlight the median and the interquartile range).** (DOCX)

**S4 Fig. Correlations between air pollutants.** (DOCX)

**S1 Table. Differences between DASH participants with complete and missing information during the study period by selected variables.** (DOCX)

**S2 Table. Selected descriptive characteristics of the DASH participants (complete cases).** (DOCX)

**S3 Table. Additional descriptive characteristics of the sample (multiply imputed).** (DOCX)

**S4 Table. Longitudinal associations between air pollutants and Blood Pressure among participants aged ll-16 years in the DASH study (Complete Cases).** (DOCX)

## Acknowledgments

We acknowledge the invaluable support of participants and their parents, the Participant Advisory Group, schools, civic leaders, local GP surgeries and community pharmacies, the Clinical Research Centre at Queen Mary University of London, the Clinical Research Facility at University College Hospital, the survey assistants, and nurses involved in DASH data collection.

## Author Contributions

**Conceptualization:** A. Karamanos, I. S. Mudway, J. K. Cruickshank, S. Harding.

**Data curation:** A. Karamanos, I. S. Mudway, F. J. Kelly, S. D. Beevers, D. Dajnak, D. Fecht, C. Elia, M. J. Maynard, J. K. Cruickshank, S. Harding.

**Formal analysis:** A. Karamanos, Y. Lu, S. Ayis, C. Elia, A. J. Webb, O. R. Molaodi, J. K. Cruickshank, S. Harding.

**Funding acquisition:** M. J. Maynard, J. K. Cruickshank, S. Harding.

**Investigation:** M. J. Maynard, J. K. Cruickshank, S. Harding.

**Methodology:** A. Karamanos, I. S. Mudway, S. Ayis, F. J. Kelly, C. Elia, A. J. Grande, S. Harding.

**Project administration:** I. S. Mudway, S. Harding.

**Resources:** J. K. Cruickshank, S. Harding.

**Software:** A. Karamanos, I. S. Mudway.

**Supervision:** J. K. Cruickshank, S. Harding.

**Visualization:** A. Karamanos.

**Writing – original draft:** A. Karamanos, S. Tandon, A. J. Webb, A. J. Grande, S. Harding.

**Writing – review & editing:** A. Karamanos, Y. Lu, I. S. Mudway, S. Ayis, F. J. Kelly, S. D. Beevers, D. Dajnak, D. Fecht, C. Elia, S. Tandon, A. J. Webb, A. J. Grande, O. R. Molaodi, M. J. Maynard, J. K. Cruickshank, S. Harding.

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
