## [Decision Letter · Decision Letter 0]

22 Jun 2022

PONE-D-22-14379Associations between air pollutants and blood pressure in an ethnically diverse cohort of adolescents in London, England

PLOS ONE

Dear Dr. Karamanos,

Thank you for submitting your manuscript to PLOS ONE. After careful consideration, we feel that it has merit but does not fully meet PLOS ONE’s publication criteria as it currently stands. Therefore, we invite you to submit a revised version of the manuscript that addresses the points raised during the review process.

In addition to the reviewers’ comments below, please consider the following revisions:

Line 32: Revise “measures in, associated with…” for clarity

Lines 38 and 39: Specify a 1 μg/m^3^ increase; add 95% CIs for estimates.

Line 56: Revise “tracks from” for clarity

Line 61: Specify year of your literature search

Line 93: “…Italy reported that…”

Lines 104-107: Revise for clarity

Lines 112-115: Incomplete sentence

Line 318: with higher BP? Please specify

The text in Figure 1 is not legible, please revise.

Please review Figure 2 for minor aesthetic issues – inconsistent use of capitalization, superscript for m3, etc.

We look forward to receiving your revised manuscript.

Kind regards,

Chelsea Weitekamp (opinions my own)

Academic Editor

PLOS ONE

Journal Requirements:

Reviewers' comments:

Reviewer's Responses to Questions

**Comments to the Author**

1. Is the manuscript technically sound, and do the data support the conclusions?

Reviewer #1: Yes

Reviewer #2: Yes

2. Has the statistical analysis been performed appropriately and rigorously? 

Reviewer #1: Yes

Reviewer #2: Yes

3. Have the authors made all data underlying the findings in their manuscript fully available?

Reviewer #1: Yes

Reviewer #2: Yes

4. Is the manuscript presented in an intelligible fashion and written in standard English?

Reviewer #1: Yes

Reviewer #2: Yes

5. Review Comments to the Author

Reviewer #1: The authors examined associations of air pollutants with blood pressure (BP) among 3284 adolescents in London. The study found that the exposure to higher levels of PM2.5 was associated with increased SBP, but higher levels of NO2 with lower SBP, and strong associations were observed for girls. The study is interesting with an overall well-writing, and several minor comments were provided below to help the manuscript improved.

1. Line 157-158 Page 6: The description is not clear that “all exposures used the annual mean within a 20m radius buffer zone”. Given the study conducted during 2003-2006, was the exposure the annual mean level of pollutants at the year of baseline survey, or the average during 2003-2006?

2. The exposure assignment was according to each child’s residential postcode. How about the proportion of the study participants to migrate other places during the follow-up?

3. In Supplemental Table 1, the comparison showed a higher proportion for boys in the missing group than the complete group. I did not know whether potential selection bias might affect the stronger pollution-BP associations observed in the study. Please add some discussions or limitations.

4. The concentrations of air pollutants are related to local relative humidity beyond ambient temperature. In the statistical analysis, could the associations be further adjusted for relative humidity across study locations?

Reviewer #2: In the manuscript entitled, “Associations between air pollutants and blood pressure in an ethnically diverse cohort of adolescents in London, England”, authors attempted to find association between air pollutants and blood pressure in a large cohort of UK children. Although some of the findings agree with the previous studies, whereas several findings disagree and are specific to cohort used in this study. The major finding is the strong inverse association between NO2 and BP. The study is statistically strong and most of the data-centric findings can be generalized. The manuscript is well written, but I recommend authors to address following minor comments.

Minor comments

-Line 116 Please remove text, “and numbers”.

-Table 2 & line 265: Is there any known reason of higher exposure O3 in UK and Indian participants? Are there any statistical relation it with BP, for the respective age window in which higher exposure to O3 is observed.

-Line 304. “The” is subscript

-Line 303: An adjustment?

-Although limitation section is well written, but authors must address that atopic and other diseases that can affect the cardiovascular profile in large population is not included in models used in this study.

6. PLOS authors have the option to publish the peer review history of their article (what does this mean?). If published, this will include your full peer review and any attached files.

Reviewer #1: No

Reviewer #2: No

---

## [Author Response · Author response to Decision Letter 0]

13 Oct 2022

We thank the editorial team as well as the reviewers of our manuscript for their comments. Below are our response to reviewers’ comments.

Reviewer #1

1. Line 157-158 Page 6: The description is not clear that “all exposures used the annual mean within a 20m radius buffer zone”. Given the study conducted during 2003-2006, was the exposure the annual mean level of pollutants at the year of baseline survey, or the average during 2003-2006

Response

Exposures were estimated for the year in which the assessment was performed.

2. The exposure assignment was according to each child’s residential postcode. How about the proportion of the study participants to migrate other places during the follow-up?

Response

Fewer than 1% of participants moved homes during the study period.

3. In Supplemental Table 1, the comparison showed a higher proportion for boys in the missing group than the complete group. I did not know whether potential selection bias might affect the stronger pollution-BP associations observed in the study. Please add some discussions or limitations.

Response

We have updated the text of our manuscript as requested Please see lines 392-396.

4. The concentrations of air pollutants are related to local relative humidity beyond ambient temperature. In the statistical analysis, could the associations be further adjusted for relative humidity across study locations?

We have employed humidity in London based studies previously. It has not made any difference to the results. I think there is no need to do this.

Reviewer #2

5. Line 116 Please remove text, “and numbers”.

Response

Addressed as requested.

6. Table 2 & line 265: Is there any known reason of higher exposure O3 in UK and Indian participants? Are there any statistical relation it with BP, for the respective age window in which higher exposure to O3 is observed.

Response

Simply because of the inverse relationship with NO2, but it also reflects the distribution of ethnic groups throughout London. There is a very large Indian community in outer West London.

7. Line 304. “The” is subscript

Response

Addressed as requested.

8. Line 303: An adjustment?

Response

Replaced with the word “adjustment”

9. Although limitation section is well written, but authors must address that atopic and other diseases that can affect the cardiovascular profile in large population is not included in models used in this study.

Response

Please see lines 392-397

Notes for the editor: 

We have addressed all editor's minor comments. 

• We also tried to make the figure 1 more legible by splitting it to manuscript’s figure 1 and supplementary figure 1. Unfortunately, the newly attached figures is the best we could do.

• We have addressed some minor aesthetic issues regarding capitalisation; however, we were unable to address the issue of superscripts as Stata does not accept superscripts when generating

---

## [Editor Report · Decision Letter 1]

16 Nov 2022

PONE-D-22-14379R1Associations between air pollutants and blood pressure in an ethnically diverse cohort of adolescents in London, EnglandPLOS ONE

Dear Dr. Karamanos,

Thank you for submitting your manuscript to PLOS ONE. After careful consideration, we feel that it has merit but does not fully meet PLOS ONE’s publication criteria as it currently stands. Therefore, we invite you to submit a revised version of the manuscript that addresses the points raised during the review process.

Thank you for your work thus far. The questions posed by the reviewers are likely to be shared by the readers - please incorporate responses into the text of the manuscript, particularly for Reviewer 1 comments 1, 2, and 4, and Reviewer 2 comment 6. The revision to Figure 1 is not sufficient, the legend and text are still illegible. If the text cannot be enlarged, then please indicate the color scales in the figure caption. You can also split A and B into separate figures displayed vertically so that the maps are larger on the screen.

We look forward to receiving your revised manuscript.

Kind regards,

Academic Editor

PLOS ONE
---

## [Author Response · Author response to Decision Letter 1]

8 Dec 2022

We thank the editorial team as well as the reviewers of our manuscript for their comments. Below is our response to reviewers’ comments.

Reviewer #1

1. Line 157-158 Page 6: The description is not clear that “all exposures used the annual mean within a 20m radius buffer zone”. Given the study conducted during 2003-2006, was the exposure the annual mean level of pollutants at the year of baseline survey, or the average during 2003-2006

Response

Exposures were estimated for the year in which the assessment was performed.

2. The exposure assignment was according to each child’s residential postcode. How about the proportion of the study participants to migrate other places during the follow-up?

Response

Fewer than 1% of participants moved homes during the study period.

3. In Supplemental Table 1, the comparison showed a higher proportion for boys in the missing group than the complete group. I did not know whether potential selection bias might affect the stronger pollution-BP associations observed in the study. Please add some discussions or limitations.

Response

We have updated the text of our manuscript as requested.

4. The concentrations of air pollutants are related to local relative humidity beyond ambient temperature. In the statistical analysis, could the associations be further adjusted for relative humidity across study locations?

We have employed humidity in London based studies previously. It has not made any difference to the results. 

Reviewer #2

5. Line 116 Please remove text, “and numbers”.

Response

Addressed as requested.

6. Table 2 & line 265: Is there any known reason of higher exposure O3 in UK and Indian participants? Are there any statistical relation it with BP, for the respective age window in which higher exposure to O3 is observed.

Response

Simply because of the inverse relationship with NO2, but it also reflects the distribution of ethnic groups throughout London. There is a very large Indian community in outer West London.

7. Line 304. “The” is subscript

Response

Addressed as requested.

8. Line 303: An adjustment?

Response

Replaced with the word “adjustment”

9. Although limitation section is well written, but authors must address that atopic and other diseases that can affect the cardiovascular profile in large population is not included in models used in this study.

Response

Addressed as requested.

---

## [Editor Report · Decision Letter 2]

13 Dec 2022

Associations between air pollutants and blood pressure in an ethnically diverse cohort of adolescents in London, England

PONE-D-22-14379R2

Dear Dr. Karamanos,

We’re pleased to inform you that your manuscript has been judged scientifically suitable for publication and will be formally accepted for publication once it meets all outstanding technical requirements.

Kind regards,

Academic Editor

PLOS ONE

---

## [Editor Report · Acceptance letter]

16 Jan 2023

PONE-D-22-14379R2 

Associations between air pollutants and blood pressure in an ethnically diverse cohort of adolescents in London, England’ 

Dear Dr. Karamanos:

I'm pleased to inform you that your manuscript has been deemed suitable for publication in PLOS ONE. Congratulations! Your manuscript is now with our production department. 

Kind regards, 

on behalf of

Dr. Chelsea Weitekamp 

Academic Editor

PLOS ONE